# Distinct ipRGC subpopulations mediate light's acute and circadian effects on body temperature and sleep

Alan C Rupp[1], Michelle Ren[2], Cara M Altimus[1], Diego C Fernandez[1†], Melissa Richardson[1], Fred Turek[2], Samer Hattar[1,3†], Tiffany M Schmidt[2*]

[1]Department of Biology, Johns Hopkins University, Baltimore, United States;
[2]Department of Neurobiology, Northwestern University, Evanston, United States;
[3]Department of Neuroscience, Johns Hopkins University, Baltimore, United States

**Abstract** The light environment greatly impacts human alertness, mood, and cognition by both acute regulation of physiology and indirect alignment of circadian rhythms. These processes require the melanopsin-expressing intrinsically photosensitive retinal ganglion cells (ipRGCs), but the relevant downstream brain areas involved remain elusive. ipRGCs project widely in the brain, including to the central circadian pacemaker, the suprachiasmatic nucleus (SCN). Here we show that body temperature and sleep responses to acute light exposure are absent after genetic ablation of all ipRGCs except a subpopulation that projects to the SCN. Furthermore, by chemogenetic activation of the ipRGCs that avoid the SCN, we show that these cells are sufficient for acute changes in body temperature. Our results challenge the idea that the SCN is a major relay for the acute effects of light on non-image forming behaviors and identify the sensory cells that initiate light's profound effects on body temperature and sleep.
DOI: https://doi.org/10.7554/eLife.44358.001

**\*For correspondence:**
tiffany.schmidt@northwestern.edu

**Present address:** †National Institute of Mental Health, Bethesda, United States

**Competing interests:** The authors declare that no competing interests exist.

## Introduction

Many essential functions are influenced by light both indirectly through alignment of circadian rhythms (photoentrainment) and acutely by a direct mechanism (sometimes referred to as 'masking') (*Mrosovsky et al., 1999*; *Altimus et al., 2008*; *Lupi et al., 2008*; *Tsai et al., 2009*; *LeGates et al., 2012*). Dysregulation of the circadian system by abnormal lighting conditions has many negative consequences, which has motivated decades of work to identify the mechanisms of circadian photoentrainment (*Golombek and Rosenstein, 2010*). In contrast, it has only recently become apparent that light exposure can also acutely influence human alertness, cognition, and physiology (*Chellappa et al., 2011*). As a result, there is a developing awareness of light quality in everyday life (*Lucas et al., 2014*). It is therefore essential to human health and society to elucidate the circuitry and coding mechanisms underlying light's acute effects.

Intriguingly, a single population of retinal projection neurons—intrinsically photosensitive retinal ganglion cells (ipRGCs)—have been implicated in the circadian and acute effects of light on many functions, including activity, sleep, and mood (*Göz et al., 2008*; *Güler et al., 2008*; *Hatori et al., 2008*; *LeGates et al., 2012*; *Fernandez et al., 2018*). ipRGCs integrate light information from rods, cones, and their endogenous melanopsin phototransduction cascade (*Schmidt et al., 2011*), and relay that light information to over a dozen central targets (*Hattar et al., 2006*; *Ecker et al., 2010*). However, the circuit mechanisms mediating ipRGC-dependent functions are largely unknown.

One notable exception is the control of circadian photoentrainment. It is accepted that ipRGCs mediate photoentrainment by direct innervation of the master circadian pacemaker, the suprachiasmatic nucleus (SCN) of the hypothalamus (*Göz et al., 2008*; *Güler et al., 2008*; *Hatori et al., 2008*;

**eLife digest** Light, whether natural or artificial, affects our everyday lives in several ways. Exposure to light impacts on our health and well-being. It plays a crucial but indirect role in helping to align our internal body clock with the 24-hour cycle of day and night, and a burst of bright light in the middle of the night can wake us up from sleep.

Decades of research have revealed the circuitry that controls the indirect effects of light on the body's internal clock. A tiny set of cells in the base of the brain called the suprachiasmatic nucleus (SCN for short) generates the body's daily or "circadian" rhythm. A small group of nerve cells in the retina of the eye called intrinsically photosensitive retinal ganglion cells (ipRGCs) connect with the SCN. These ipRGCs relay information about light to the SCN to ensure that daily rhythms happen at the appropriate times of day. But scientists do not yet know if the same brain circuits regulate the direct effects of light on alertness.

Mice are often used in studies of circadian rhythms but, unlike humans, mice are normally active at night and sleep throughout the day. This means that a burst of bright light in the middle of the night causes mice to become less alert.

Now, in experiments with mice, Rupp et al. show there are two separate circuits from the retina to the brain that influence wakefulness. In the experiments, some mice were genetically engineered to only have ipRGCs that connect with the SCN and to lack those that connect with other brain areas. These mice lived in cages with a normal day/night cycle and their body temperature and sleep-related brain activity were monitored as Rupp et al. sporadically exposed them to bright light at night. These mice continued their normal routines and were unaffected by the bursts of light. In a second set of experiments, ipRGCs that do not connect with the SCN were activated in other mice. This caused an immediate and sustained drop in the body temperature of the mice, which is linked to them becoming less alert.

The experiments suggest that the circuit that connects ipRGCs to the SCN to align the body's circadian rhythm with light does not control the direct effect of light on wakefulness. Instead, a separate circuit that extends from ipRGCs to an unknown part of the brain area influences wakefulness. Better understanding this second circuit could allow scientists to develop ways to keep people like emergency personnel or overnight shift workers awake and alert at night while avoiding harmful disruptions to their circadian rhythms.

DOI: https://doi.org/10.7554/eLife.44358.002

*Jones et al., 2015*). This is supported by studies demonstrating that genetic ablation of ipRGCs results in mice with normal circadian rhythms that 'free-run' with their endogenous rhythm, independent of the light/dark cycle (*Göz et al., 2008*; *Güler et al., 2008*; *Hatori et al., 2008*). Further, mice with genetic ablation of all ipRGCs except those that project to the SCN and intergeniculate leaflet (IGL) display normal circadian photoentrainment (*Chen et al., 2011*), suggesting that ipRGC projections to the SCN/IGL are sufficient for photoentrainment.

In comparison, the mechanisms by which ipRGCs mediate acute light responses remain largely a mystery. Genetic ablation of ipRGCs or their melanopsin phototransduction cascade blocks or attenuates the acute effects of light on sleep (*Altimus et al., 2008*; *Lupi et al., 2008*; *Tsai et al., 2009*), wheel-running activity (*Mrosovsky and Hattar, 2003*; *Güler et al., 2008*), and mood (*LeGates et al., 2012*; *Fernandez et al., 2018*). This dual role of ipRGCs in circadian and acute light responses suggests they may share a common circuit mechanism. However, whether the circuit basis for ipRGCs in the acute effects of light and circadian functions is through common or divergent pathways has yet to be determined. ipRGCs project broadly in the brain beyond the SCN (*Hattar et al., 2002*; *Hattar et al., 2006*; *Gooley et al., 2003*; *Baver et al., 2008*). Additionally, ipRGCs are comprised of multiple subpopulations with distinct genetic, morphological, and electrophysiological signatures (*Baver et al., 2008*; *Schmidt and Kofuji, 2009*; *Ecker et al., 2010*; *Schmidt et al., 2011*) and distinct functions (*Chen et al., 2011*; *Schmidt et al., 2014*). Though there are rare exceptions (*Chen et al., 2011*; *Schmidt et al., 2014*), the unique roles played by each ipRGC subsystem remain largely unknown.

It is currently unknown whether distinct ipRGC subpopulations mediate both the acute and circadian effects of light, and two major possibilities exist for how this occurs: (1) ipRGCs mediate both acute and circadian light responses through their innervation of the SCN or (2) ipRGCs mediate circadian photoentrainment through the SCN, but send collateral projections elsewhere in the brain to mediate acute light responses. To date, the predominant understanding has centered on a role for the SCN in both acute and circadian responses to light (*Muindi et al., 2014*; *Morin, 2015*; *Bedont et al., 2017*). However, this model has been controversial due to complications associated with SCN lesions (*Redlin and Mrosovsky, 1999*) and alternative models proposing a role for direct ipRGC input to other central targets (*Redlin and Mrosovsky, 1999*; *Lupi et al., 2008*; *Tsai et al., 2009*; *Hubbard et al., 2013*; *Muindi et al., 2014*). Here, we sought to address the question of how environmental light information—through ipRGCs—mediates both the circadian and acute regulation of physiology. To do so, we investigated the ipRGC subpopulations and coding mechanisms that mediate body temperature and sleep regulation by light. We find that a molecularly distinct subset of ipRGCs is required for the acute, but not circadian, effects of light on thermoregulation and sleep. These findings suggest that, contrary to expectations, functional input to the SCN is not sufficient to drive the acute effects of light on these behaviors. These findings provide new insight into the circuits through which light regulates behavior and physiology.

## Results

### Brn3b-positive ipRGCs are required for light's acute effects on thermoregulation

To identify mechanisms of acute thermoregulation, we maintained mice on a 12 hr/12 hr light/dark cycle and then presented a 3 hr light pulse two hours into the night (Zeitgeber time 14, ZT14) while measuring core body temperature (*Figure 1A*). The nocturnal light pulse paradigm is well-established for studying acute regulation of sleep and wheel-running activity (*Mrosovsky et al., 1999*; *Mrosovsky and Hattar, 2003*; *Altimus et al., 2008*; *Lupi et al., 2008*). We focused first on body temperature because of its critical role in cognition and alertness (*Wright et al., 2002*; *Darwent et al., 2010*), sleep induction and quality (*Kräuchi et al., 1999*), metabolic control (*Kooijman et al., 2015*), and circadian resetting (*Buhr et al., 2010*).

Body temperature photoentrains to the light/dark cycle with peaks during the night and troughs during the day (*Figure 1B*). Both rodents and humans utilize ocular light detection to acutely adjust body temperature in response to a nocturnal light pulse (*Dijk et al., 1991*; *Cajochen et al., 2005*), though how this body temperature change is initiated by the retina and relayed to the brain is unknown. When we presented wildtype mice with a nocturnal light pulse, we observed a decrease in both body temperature and general activity compared to the previous night (*Figure 1C*). The decrease in body temperature and activity was sustained for the entire 3 hr stimulus, with moderate rundown (*Figure 1C*).

We observed that acute body temperature regulation only occurred at relatively bright light intensities (>100 lux) (*Figure 1—figure supplement 1*). This, in combination with previous reports that body temperature regulation is most sensitive to short-wavelength light (*Cajochen et al., 2005*), suggested that it might be mediated by the insensitive and blue-shifted melanopsin phototransduction (*Lucas et al., 2001*; *Do et al., 2009*). To test this, we measured body temperature in mice lacking either functional rods and cones (melanopsin-only: $Gnat1^{-/-}$; $Gnat2^{-/-}$) or lacking melanopsin (melanopsin KO: $Opn4^{-/-}$). Both genotypes photoentrained their body temperature (*Figure 1D,E*), with an amplitude indistinguishable from wildtype (*Figure 1F*). However, we found that acute body temperature decrease to a nocturnal light pulse was present in melanopsin-only mice ($Gnat1^{-/-}$; $Gnat2^{-/-}$) (*Figure 1G,H* and *Figure 1—figure supplement 2*), but absent from melanopsin knockout mice ($Opn4^{-/-}$) (*Figure 1I,J* and *Figure 1—figure supplement 2*). This indicates that melanopsin is critical for light's ability to drive acute body temperature decreases, as it is for acute sleep induction (*Altimus et al., 2008*; *Lupi et al., 2008*; *Tsai et al., 2009*). These results suggest that ipRGCs are the only retinal cells that are necessary and sufficient for acute thermoregulation by light.

ipRGCs comprise multiple subtypes (M1-M6) with distinct gene expression profiles, light responses, and central projections (*Schmidt et al., 2011*; *Quattrochi et al., 2019*), prompting us to

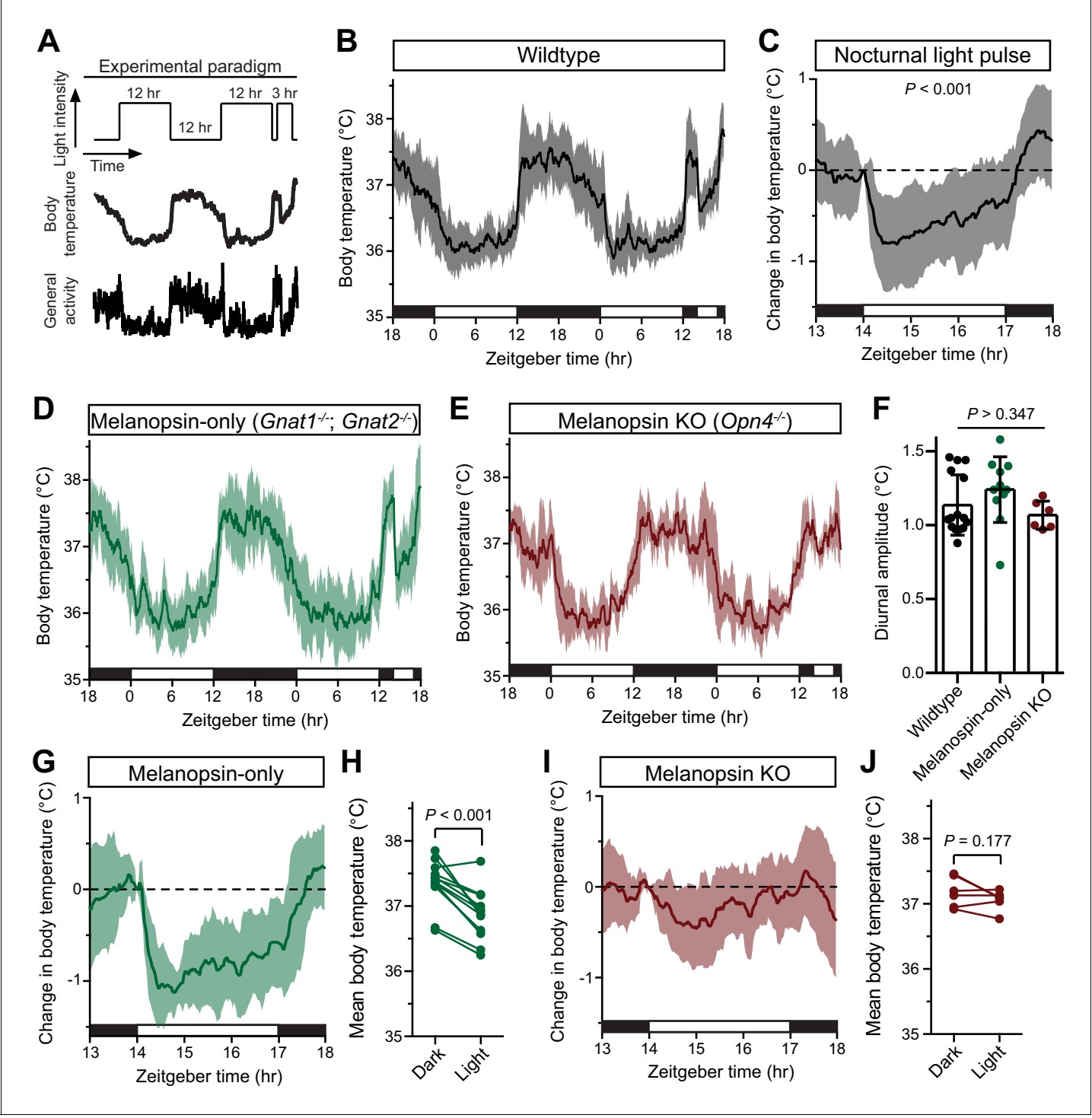

**Figure 1.** Melanopsin mediates the acute effects of light on body temperature. (**A**) Paradigm to measure body temperature continuously in a 12:12 light dark cycle with a 3 hr light pulse at ZT14. (**B**) 48 hr of continuous body temperature monitoring in wildtype male mice (n = 13) (**C**) Relative body temperature in WT during light pulse, compared to baseline (ZT14). p<0.001, paired t-test of mean temperature compared to previous night. (**D**) Melanopsin-only mice (*Gnat1*$^{-/-}$; *Gnat2*$^{-/-}$, n = 11) and (**E**) melanopsin knockout (*Opn4*$^{-/-}$, n = 6) 48 hr diurnal body temperature. (**F**) Diurnal body temperature amplitude in the three groups. p>0.347 for effect of group by one-way ANOVA. (**G**) Body temperature in melanopsin-only during light pulse, relative to baseline (ZT14). (**H**) Paired comparison of mean body temperature during light pulse compared to previous night. p<0.001 by paired t-test. (**I**) Body temperature in melanopsin knockout during light pulse, relative to baseline (ZT14). (**J**) Paired comparison of mean body temperature during light pulse compared to previous night. All summarized data are mean ± standard deviation.

*Figure 1 continued on next page*

*Figure 1 continued*

DOI: https://doi.org/10.7554/eLife.44358.003

The following source data and figure supplements are available for figure 1:

**Source data 1.** Temperature data for *Figure 1*.

DOI: https://doi.org/10.7554/eLife.44358.006

**Figure supplement 1.** Intensity-dependent decrease in core body temperature during a nocturnal light pulse.

DOI: https://doi.org/10.7554/eLife.44358.004

**Figure supplement 1—source data 1.** Temperature data for *Figure 1—figure supplement 1*.

DOI: https://doi.org/10.7554/eLife.44358.007

**Figure supplement 2.** Melanopsin-dependence of light-induced body temperature changes.

DOI: https://doi.org/10.7554/eLife.44358.005

ask which subtypes mediate acute thermoregulation. ipRGCs can be molecularly subdivided based on whether they express the transcription factor Brn3b. Brn3b(+) ipRGCs project to many structures including the olivary pretectal nucleus (OPN) and dorsal lateral geniculate nucleus (dLGN), but largely avoid the SCN (*Chen et al., 2011*; *Li and Schmidt, 2018*). In contrast, Brn3b(–) ipRGCs project extensively to the SCN and intergeniculate leaflet (IGL), while avoiding the OPN and dLGN (*Chen et al., 2011*). Non-M1 (i.e. M2-M6) ipRGC subtypes express Brn3b, along with the majority of M1 ipRGCs. Interestingly, just 200 (out of 700–800) M1 ipRGCs lack any Brn3b expression (*Chen et al., 2011*). Ablation of Brn3b(+) ipRGCs using melanopsin-Cre and a Cre-dependent diphtheria toxin knocked into the *Brn3b* locus (Brn3b-DTA: $Opn4^{Cre/+};Brn3b^{zDTA/+}$) removes virtually all ipRGC input to brain areas aside from the SCN and IGL (*Chen et al., 2011*; *Li and Schmidt, 2018*), and these mice lack a pupillary light reflex and show deficits in contrast sensitivity, but retain circadian photoentrainment of wheel-running activity (*Chen et al., 2011*; *Schmidt et al., 2014*).

When we measured body temperature in Brn3b-DTA mice, we found that their body temperature was photoentrained with a similar amplitude to controls (*Figure 2A–C*). However, despite the presence of melanopsin in the Brn3b(-) ipRGCs of Brn3b-DTA mice ($Opn4^{Cre/+};Brn3b^{zDTA/+}$), they did not acutely decrease body temperature in response to a nocturnal light pulse (*Figure 2F,G*). Importantly, melanopsin heterozygous littermate controls ($Opn4^{Cre/+}$) displayed normal acute thermoregulation by light (*Figure 2D,E*), indicating that halving melanopsin gene dosage is not the cause of the impaired body temperature decrease in Brn3b-DTA mice. Additionally, when we compared the change in body temperature of Control to Brn3b-DTA mice during that light pulse, we found that Control mice showed a significantly larger decrease in body temperature (*Figure 2—figure supplement 1*). These results demonstrate that Brn3b(+) ipRGCs are required for acute thermoregulation by light but not photoentrainment of body temperature and reveal that light information to the SCN is sufficient for circadian photoentrainment of body temperature, but not its acute regulation.

## Brn3b-positive ipRGCs are sufficient for acute thermoregulation

Our data thus far suggest that there are two functionally distinct populations of ipRGCs that regulate thermoregulation: (1) Brn3b(–) ipRGCs that project to the SCN to mediate circadian photoentrainment of body temperature and (2) Brn3b(+) ipRGCs that project elsewhere in the brain and are necessary to mediate acute thermoregulation. If Brn3b(+) ipRGCs are not just necessary, but also sufficient, for acute thermoregulation, then activation of this population at ZT14 should result in a body temperature decrease. To test if Brn3b(+) ipRGCs are sufficient for acute thermoregulation, we expressed a chemogenetic activator in Brn3b(+) RGCs (*Figure 3A*, $Brn3b^{Cre/+}$ with intravitreal AAV2-hSyn-DIO-hM3Dq-mCherry, we refer to these mice as Brn3b-hM3Dq). As a control, we also injected this virus into Control ($Brn3b^{+/+}$) littermates. We then injected both genotypes first with PBS at ZT14 on the first night, and CNO at ZT14 on the second night. This technique allowed for statistical within animal comparisons of body temperature changes in response to PBS versus CNO injection. Importantly, CNO did not cause a significant decrease in body temperature in the absence of hM3Dq (*Figure 3—figure supplement 1*). This technique allowed us to acutely activate the Brn3b(+) RGCs with the DREADD agonist clozapine N-oxide (CNO) (*Armbruster et al., 2007*). We found that after intravitreal viral delivery, many RGCs were infected, including melanopsin-expressing ipRGCs (*Figure 3A* and *Figure 3—figure supplement 1*).

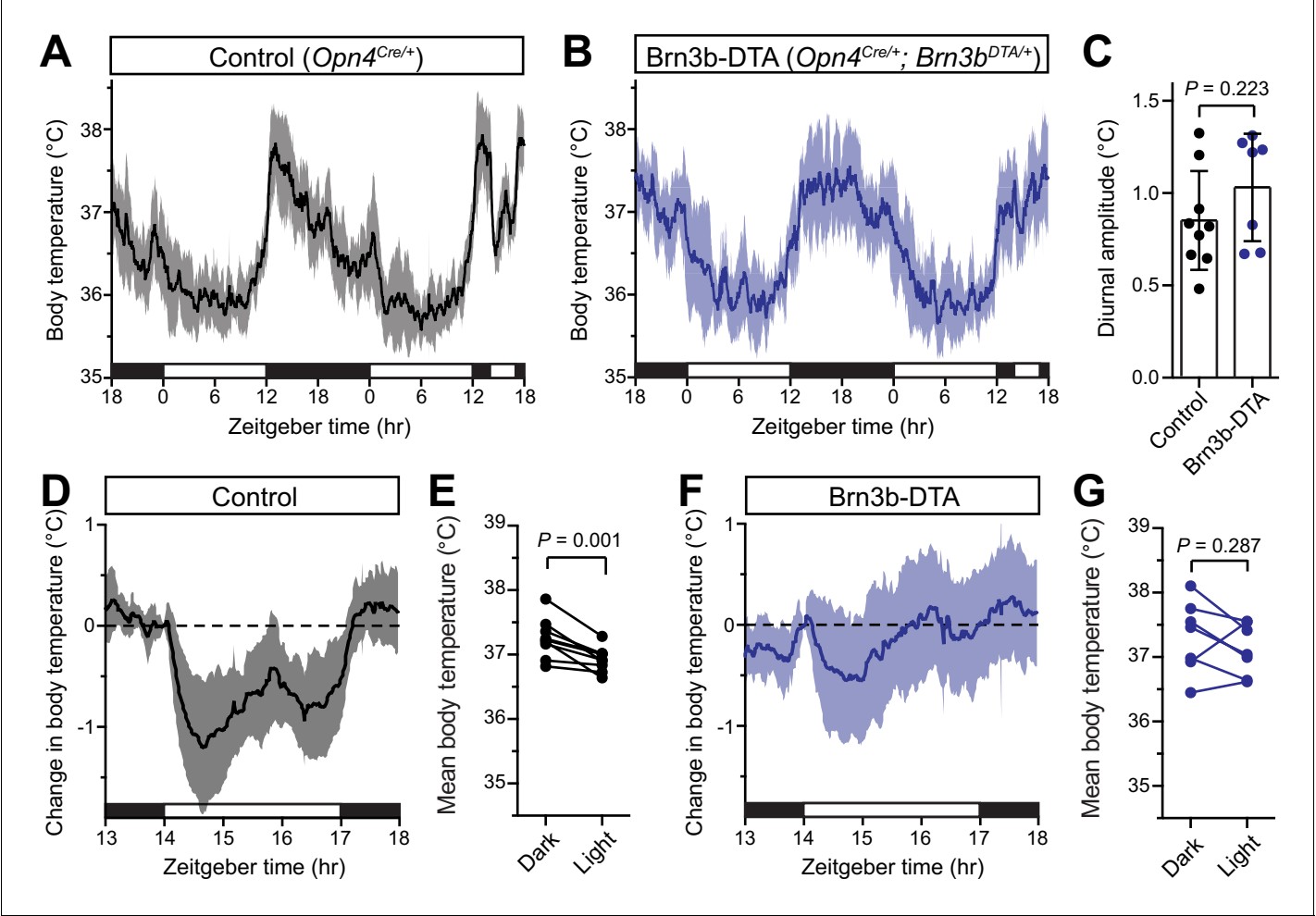

**Figure 2.** Brn3b-negative ipRGCs are insufficient for acute body temperature regulation via the SCN. (A) Diurnal body temperature in control (*Opn4^{Cre/+}*, n = 9) and (B) Brn3b-DTA (*Opn4^{Cre/+};Brn3b^{DTA/+}*, n = 7). (C) Diurnal body temperature amplitude in the two groups. p=0.223 by t-test. (D) Body temperature in control during light pulse, relative to baseline (ZT14). (E) Paired comparison of mean body temperature during light pulse compared to previous night. p=0.001 by paired t-test. (F) Body temperature in Brn3b-DTA during light pulse, relative to baseline (ZT14). (G) Paired comparison of mean body temperature during light pulse compared to previous night. p=0.287 by paired t-test. All summarized data are mean ± standard deviation.

DOI: https://doi.org/10.7554/eLife.44358.008

The following source data and figure supplement are available for figure 2:

**Source data 1.** Temperature data for *Figure 2*.
DOI: https://doi.org/10.7554/eLife.44358.010
**Figure supplement 1.** Brn3b-DTA body temperature regulation with light.
DOI: https://doi.org/10.7554/eLife.44358.009

The body temperature of Brn3b-hM3Dq mice photoentrained to a normal light/dark cycle (*Figure 3B*). Following CNO administration to Brn3b-hM3Dq mice at ZT14 to depolarize Brn3b(+) RGCs, we observed a robust decrease in body temperature that lasted at least 6 hr (*Figure 3D*). Importantly, PBS administration in Brn3b-hM3Dq mice (*Figure 3C*) and nocturnal CNO administration in wildtype control mice (*Figure 3—figure supplement 2*) had no measurable effect on body temperature, while CNO administration significantly decreased body temperature in Brn3b-hM3Dq compared to pre-injection temperature (*Figure 3—figure supplement 2*). Together, these results demonstrate that Brn3b(+) ipRGCs mediate the acute effects of light on body temperature though extra-SCN projection(s), while Brn3b(–) ipRGCs mediate circadian photoentrainment of body temperature by projections to the SCN and/or IGL.

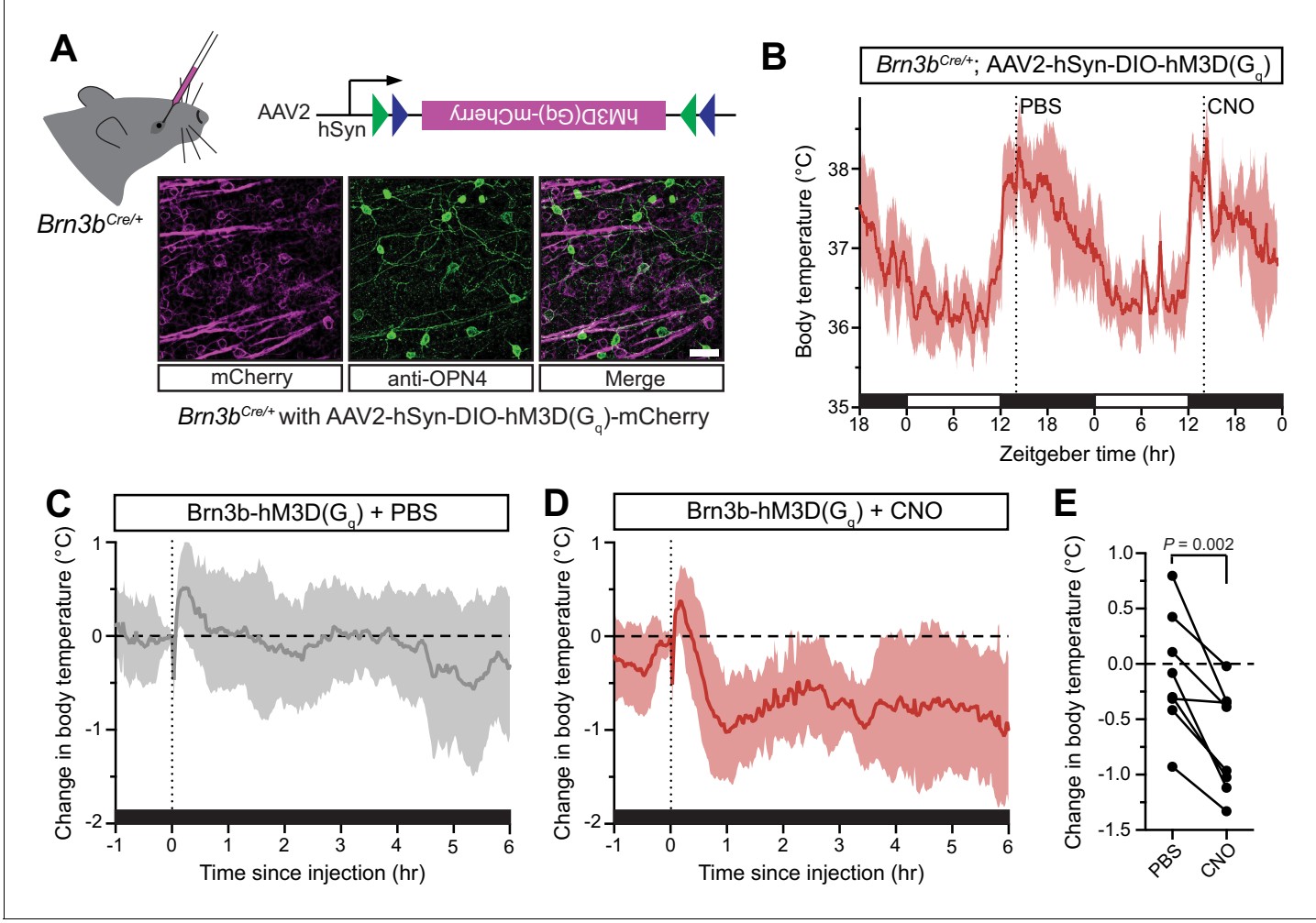

**Figure 3.** Activation of Brn3b-positive RGCs is sufficient to drive sustained body temperature decreases. (**A**) Diagram of intravitreal delivery of AAV2-hSyn-DIO-hM3Dq-mCherry to *Brn3b^{Cre/+}* mice, and confirmation of infection of ipRGCs. (**B**) 54 hr continuous diurnal body temperature recordings in Brn3b-hM3Dq mice, with injections of PBS then CNO on consecutive nights at ZT14. (**C**) Change in body temperature after PBS injection, relative to baseline (time of injection). (**D**) Change in body temperature after CNO injection, relative to baseline (time of injection). (**E**) Paired comparison of the change in body temperature with either PBS or CNO injection, compared to temperature at injection time. p=0.002 by paired t-test. All summarized data are mean ± standard deviation.

DOI: https://doi.org/10.7554/eLife.44358.011

The following source data and figure supplements are available for figure 3:

**Source data 1.** Temperature data for *Figure 3*.
DOI: https://doi.org/10.7554/eLife.44358.014
**Figure supplement 1.** Brn3b-Cre::hM3D(G_q) expression and control experiments.
DOI: https://doi.org/10.7554/eLife.44358.012
**Figure supplement 1—source data 1.** Temperature data for *Figure 3—figure supplement 1*.
DOI: https://doi.org/10.7554/eLife.44358.015
**Figure supplement 2.** No effect of CNO on body temperature in wildtype mice.
DOI: https://doi.org/10.7554/eLife.44358.013

## Brn3b-positive ipRGCs are required for light's acute effects on sleep

We next examined the contribution of Brn3b(+) and Brn3b(-) ipRGCs to sleep. To do this, we used EEG and EMG recordings to compare the sleep behavior of Control (*Opn4^{Cre/+}*) and Brn3b-DTA mice. We first analyzed the daily sleep patterns and proportion of rapid eye movement (REM) and non-REM (NREM) sleep in Control and littermate Brn3b-DTA animals. We found that Brn3b-DTA mice show normal photoentrainment of sleep and similar percent time of sleep across the 24 hour

day, with only one 30 min bin at ZT12 (light offset) showing a significant difference between Control and Brn3b-DTA animals (*Figure 4A,B*). This is consistent with previous reports of normal circadian photoentrainment of daily activity rhythms in Brn3b-DTA mice (*Chen et al., 2011*). Control and Brn3b-DTA mice also showed similar total percent time awake or asleep across an entire day (*Figure 4C*), though Brn3b-DTA mice showed a small, but significant, increase in the proportion of total sleep that was classified as NREM and decrease in the proportion of total sleep that was classified as REM (*Figure 4—figure supplement 1A*).

We hypothesized that this small difference in sleep at lights-off in Brn3b-DTA mice could be due to a defect in their acute response to light for sleep modulation. To test this, we subjected mice to a 3 hr light pulse from ZT14–17 (*Altimus et al., 2008*), when the homeostatic drive for sleep is low and Control and Brn3b-DTA animals display similar amounts of sleep (*Figure 4A,B*). We found that in Control mice, a light pulse decreased time awake and increased time asleep relative to baseline (previous day) (*Figure 4C,D*), while in Brn3b-DTA mice a light pulse caused no change in total percent time asleep or awake (*Figure 4F,G*), but moderately increased sleep in the first 30 min bin (*Figure 4F*). Importantly, when we compared the time spent asleep during the light pulse between control and Brn3b-DTA animals, the control mice slept significantly more (*Figure 4—figure supplement 2*). Neither Control nor Brn3b-DTA animals showed any change in proportion of non-REM or REM sleep in response to the light pulse (*Figure 4—figure supplement 1B,C*). These data show that Brn3b(+) ipRGCs are necessary for the acute light induction of sleep. Consistent with our body temperature data, although Brn3b-DTA mice have apparently normal input to the SCN and show normal circadian photoentrainment of wheel-running activity (*Chen et al., 2011*), body temperature (*Figure 2*), and sleep (*Figure 4*), this ipRGC innervation of the SCN is not sufficient to drive the normal light induction of sleep. These disruptions in light's acute effects on thermoregulation and sleep are circuit specific effects because Brn3b-DTA mice showed robust inhibition of wheel running behavior to a 3 hr light pulse delivered from ZT14-17 (*Figure 4—figure supplement 3*).

## Discussion

We show here that for the same physiological outcome, the acute effects of light are relayed through distinct circuitry from that of circadian photoentrainment, despite both processes requiring ipRGCs. Our results suggest that for thermoregulation and sleep, ipRGCs can be genetically and functionally segregated into Brn3b(+) 'acute' cells, and Brn3b(–) 'circadian' cells. Because Brn3b(+) cells largely avoid the SCN, and Brn3b(–) cells preferentially target the SCN, our results discount a critical role for the SCN in acute light responses in these two behaviors, and instead implicate direct ipRGC projections to other brain areas (*Gooley et al., 2003*; *Hattar et al., 2006*). Surprisingly, Brn3b(-) cells are sufficient to drive the acute and circadian effects of light on wheel running activity, demonstrating further divergence in the circuits mediating the acute effects of light on behavior, and suggesting that, unlike for thermoregulation and sleep, acute and circadian regulation of activity is driven via the SCN.

Our results indicate that activation of Brn3b(+) RGCs at ZT14 using the Brn3b[Cre] line in combination with Gq-DREADDs is sufficient to induce a body temperature decrease. Because other (non-ipRGC) RGC types express Brn3b (*Badea et al., 2009*), this manipulation likely also activated multiple non-ipRGCs in addition to Brn3b(+) ipRGCs. However, our data indicate that melanopsin signaling (*Figure 1*), and therefore ipRGCs, are required for the acute effects of light on thermoregulation. Moreover, when we ablate Brn3b(+) ipRGCs, this acute effect of light on thermoregulation is also lost (*Figure 2*), again arguing for a necessity of ipRGCs for this behavior. Therefore, though we are unable to specifically activate only Brn3b(+) ipRGCs using available genetic tools, we think it highly likely that the temperature changes driven by the activation of all Brn3b(+) RGCs is occurring through ipRGCs.

The specific Brnb(+) ipRGC subtypes that mediate the light's acute effects on body temperature and sleep remain a mystery. A majority of all known ipRGC subtypes (M1–M6) are lost in Brn3b-DTA mice (*Chen et al., 2011*), with the exception of a subset of ~200 M1 ipRGCs. In agreement with this, ipRGC projections to all minor hypothalamic targets are lost in Brn3b-DTA mice, while innervation of the SCN and part of the IGL remains intact (*Chen et al., 2011*; *Li and Schmidt, 2018*). This suggests that all non-M1 subtypes and a portion of M1 ipRGCs are Brn3b(+). Each subtype has a distinct reliance on melanopsin versus rod/cone phototransduction for light detection (*Schmidt and Kofuji,*

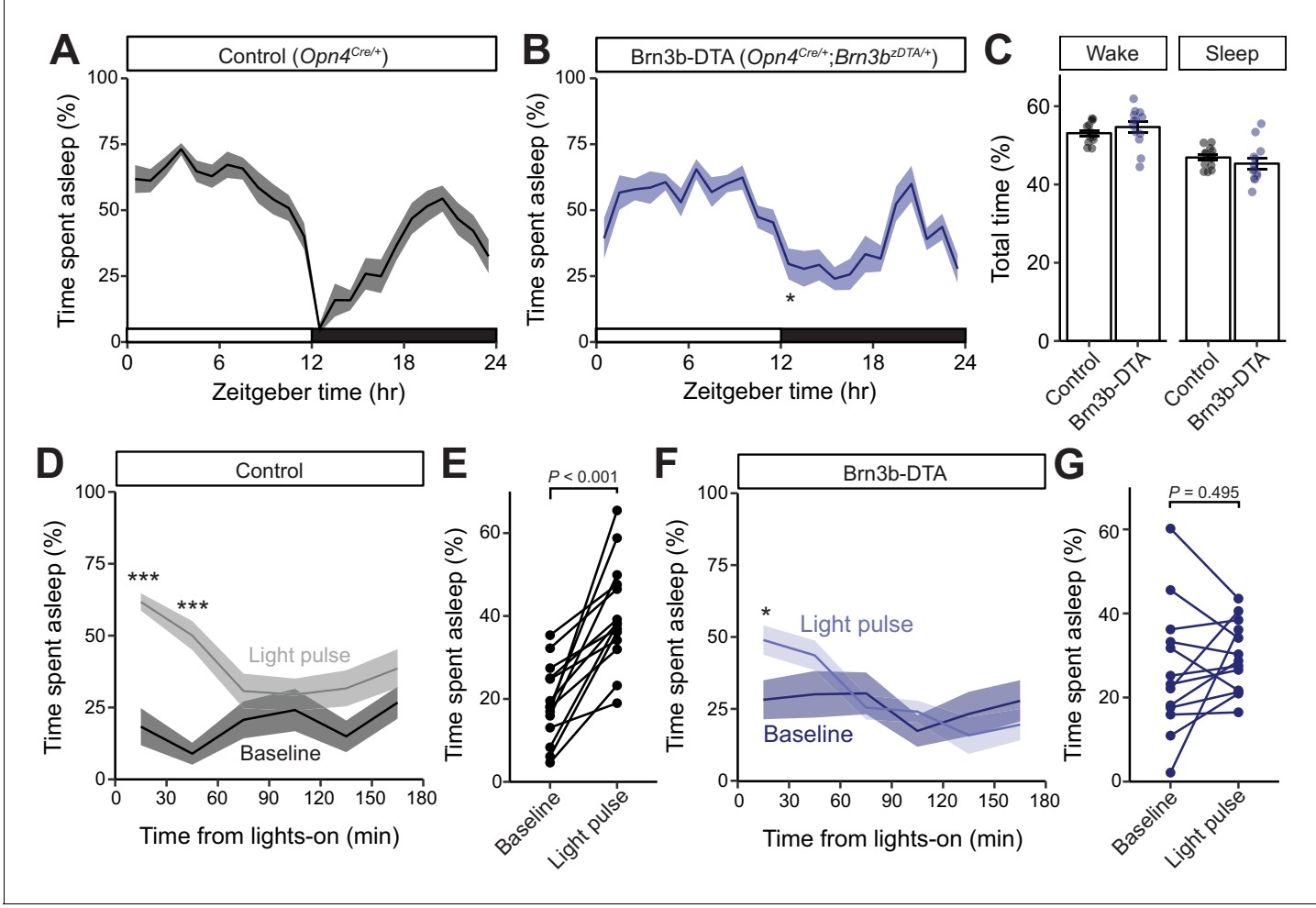

**Figure 4.** Brn3b-positive ipRGCs are not required for circadian photoentrainment of sleep, but are required for its acute induction by light. (A–C) Percent time spent asleep in 1 hr bins across the 24 hr day for (A) Control (black) mice (n = 14) and (B) Brn3b-DTA (blue) mice (n = 13) lacking Brn3b-positive ipRGCs. Both lines showed normal photoentrainment of sleep, with no main effect of genotype compared to Control by repeated-measures two-way ANOVA (F (1, 25)=1.108, p=0.303). Brn3b-DTA mice showed a significant reduction in sleep only at lights off (ZT 12) by Sidak's multiple comparisons test (p=0.029). (C) Percent time spent awake and asleep in Control (black) and Brn3b-DTA mice (blue). No differences were observed between genotypes by t-test (p=0.316). (D–G) Percent time spent asleep for (D) Control mice (black) and (F) Brn3b-DTA mice (blue) at baseline (dark line) and during the three hour light pulse (light line). Significant difference from baseline determined by repeated measures two-way ANOVA. Significant effect of treatment for Controls (F (1, 13)=38.09, p<0.001), but not for Brn3b-DTA (F (1, 12)=0.8496, p=0.375). (E) Control mice show significantly more sleep and less wake during a light pulse (paired t-test) while (G) Brn3b-DTA mice showed no change in percent sleep or wake during the same period. Data are mean for ZT14–17. All summarized data are mean ± SEM.

DOI: https://doi.org/10.7554/eLife.44358.016

The following source data and figure supplements are available for figure 4:

**Source data 1.** Sleep data for *Figure 4*.
DOI: https://doi.org/10.7554/eLife.44358.020
**Figure supplement 1.** NREM and REM measurements in Control and Brn3b-DTA mice.
DOI: https://doi.org/10.7554/eLife.44358.017
**Figure supplement 2.** Brn3b-DTA body temperature regulation with light.
DOI: https://doi.org/10.7554/eLife.44358.018
**Figure supplement 3.** Wheel-running activity in Brn3b-DTA mice.
DOI: https://doi.org/10.7554/eLife.44358.019

*2009*). The necessity and sufficiency of melanopsin in mediating acute effects of light on body temperature (*Figure 1*) and sleep (*Altimus et al., 2008*; *Lupi et al., 2008*; *Tsai et al., 2009*) suggests that a subtype with strong melanopsin, but weak rod/cone photodetection is responsible – possibly either M1 or M2 cells. However, experiments to tease this apart will require novel methods to specifically manipulate ipRGC subtypes that are currently unavailable.

The brain areas that mediate the acute effects of light on physiology are essentially unknown. There are many candidate areas that both receive direct ipRGC innervation and have been documented to be involved in light's acute effects on physiology, including the preoptic areas (*Muindi et al., 2014*), the ventral subparaventricular zone (*Kramer et al., 2001*), and the pretectum/ superior colliculus (*Miller et al., 1998*). Aside from the SCN, ipRGC projections to the median (MPO) and ventrolateral preoptic (VLPO) areas have been the most widely supported. The preoptic areas are involved in sleep and body temperature regulation (*Szymusiak and McGinty, 2008*; *Nakamura, 2011*) and are activated by an acute light pulse (*Lupi et al., 2008*; *Tsai et al., 2009*). In support of our behavioral findings, ipRGC projections to each of these areas is lost in Brn3b-DTA animals (*Li and Schmidt, 2018*). However, ipRGC projections to these areas are sparse (*Gooley et al., 2003*; *Hattar et al., 2006*), suggesting their activation by light may be indirect.

In contrast, the superior colliculus (SC) and pretectum receive robust innervation from ipRGCs (*Hattar et al., 2002*; *Hattar et al., 2006*; *Gooley et al., 2003*; *Ecker et al., 2010*), their lesioning blocks light's acute effects on sleep (*Miller et al., 1998*), and melanopsin knockout mice lose light-induced cFOS expression in the SC (*Lupi et al., 2008*). However, the SC and pretectum receive robust innervation from many conventional RGCs, making the requirement for melanopsin and ipRGCs in acute sleep and body temperature regulation difficult to reconcile. It is also possible (and perhaps probable), that multiple ipRGC target regions are involved, with distinct areas mediating distinct physiological responses. Future studies silencing each retinorecipient target while activating Brn3b(+) ipRGCs will be necessary to tease apart the downstream circuits mediating light's acute effects on physiology.

Alternatively, it remains possible that direct ipRGC control of body temperature is the primary and critical step for many acute responses to light that are mediated by ipRGCs. In support of this possibility, changes in body temperature and heat loss can directly influence sleep induction (*Kräuchi et al., 1999*). This change in sleep is in turn presumably causative of at least some of light's effects on wheel-running and general activity (*Mrosovsky et al., 1999*). Further, core body temperature can acutely regulate cognition and alertness (*Wright et al., 2002*; *Darwent et al., 2010*). It is therefore possible that ipRGCs can have widespread influence on an animal's basic physiology and cognitive function simply by regulating body temperature.

Together, our identification of the photopigment and the retinal circuits mediating acute body temperature and sleep induction will facilitate better methods to promote or avoid human alertness and cognition at appropriate times of day (*Chellappa et al., 2011*). Our results support many recent efforts to capitalize on the specific light-detection properties of melanopsin (*Lucas et al., 2014*), such as its insensitivity and short-wavelength preference, to promote or avoid its activation at different times of day. However, this approach is problematic because acute activation of melanopsin to promote alertness has the unintended effect of shifting the circadian clock (*Provencio et al., 1994*), thereby making subsequent sleep difficult. Our identification that the Brn3b(+) ipRGCs specifically mediate light's acute effects on body temperature provides a cellular basis for developing targeted methods for promoting acute alertness, while minimizing circadian misalignment.

## Materials and methods

**Key resources table**

| Reagent type (species) or resource | Designation | Source or reference | Identifiers |
|---|---|---|---|
| Genetic reagent (*Mus musculus*) | Opn4$^{tauLacZ}$ | *Hattar et al., 2002* | Jax: 021153 RRID:MGI:5520170 |

*Continued on next page*

*Continued*

| Reagent type (species) or resource | Designation | Source or reference | Identifiers |
|---|---|---|---|
| Genetic reagent (*Mus musculus*) | Gnat1$^{-/-}$ | PMID: 11095744 | |
| Genetic reagent (*Mus musculus*) | Gnat2$^{Cpfl3}$ | PMID: 17065522 | Jax: 006795 |
| Genetic reagent (*Mus musculus*) | Opn4$^{Cre}$ | *Ecker et al., 2010* | RRID:MGI:5285910 |
| Genetic reagent (*Mus musculus*) | Brn3b$^{zDTA}$ | *Chen et al., 2011* | RRID:MGI:5285910 |
| Genetic reagent (*Mus musculus*) | Brn3b$^{Cre}$ | PMID: 24608965 | RRID:IMSR_JAX:030357 |
| Antibody | anti-OPN4 (rabbit polyclonal) | Advanced Targeting Systems | AB-N38 (1:1000) RRID:AB_1608077 |
| Antibody | AlexaFluor 488, anti-rabbit (goat polyclonal) | Life Technologies | A-11008 (1:1000) RRID:AB_143165 |
| Viral reagent | AAV2-hSyn-DIO-hM3Dq-mCherry | UNC Vector Core | |
| Chemical compound, drug | Clozpine-N-oxide | Sigma | |
| Software | R 3.5.2 | https://cran.r-project.org/ | |
| Software | Graphpad Prism 7.0 | https://www.graphpad.com/scientific-software/prism/ | |

## Animals (body temperature)

All procedures were conducted in accordance with NIH guidelines and approved by the Institutional Animal Care and Use Committee of Johns Hopkins University. All mice were maintained on a mixed C57Bl/6J; 129Sv/J background and kept on ad libitum food and water under a 12 hr/12 hr light/dark cycle in group housing until experimentation, with temperature and humidity control. Male and female mice between the ages of 2 and 6 months were used for analysis.

## Body temperature recordings

Each mouse was single-housed at the time of experiment. Surgery was conducted under tribromoe-thanol (Avertin) anestheshia and a telemetric probe (Starr G2 E-Mitter) was implanted in the perito-neal cavity to monitor core body temperature and general activity. Data were collected in continuous 1- or 2 min bins using VitalVIEW software and analyzed in Microsoft Excel. All experiments were conducted at least 10 days after surgery. Lights were controlled by a programmable timer and all lights were 6500K CFL bulbs illuminated each cage at ~500 lux. Light intensity (*Figure 1—figure supplement 1*) was modulated using neutral density filters (Roscolux).

*Brn3b*$^{Cre/+}$ or Control littermate mice were anesthetized with tribromoethanol (Avertin) and 0.5–1 µl AAV2-hSyn-DIO-hM3Dq-mCherry (UNC Vector Core) was injected intravitreally in one eye using a picospritzer and pulled pipet. At least one week later, animals underwent surgery for implantation of telemetric probes (as above). All experiments were conducted at least 10 days after telemetric probe implantation and at least three weeks after viral injection. After behavior, the eyes of each animal were inspected to ensure that >50% infection had been achieved (assessed by fluorescence detectable across more than half of the retina). We routinely saw >80% of the retinas were infected as we have described previously (*Keenan et al., 2016*).

Diurnal amplitude was measured by subtracting the mean body temperature for the light cycle (ZT0-12) from the mean body temperature for the dark cycle (ZT12-24). Mean body temperature during testing used all data from ZT14-17. Comparisons were performed in one of two ways. First, we compared the mean body temperature during this period on the control (dark) night to that on the night where the light pulse was given. Additionally, we compared the change in body temperature between ZT14 (which served as a baseline) and the mean body temperature from ZT14-17

between the control night and the night where the light pulse was given. For CNO experiments, injections were carried out near ZT14, but specific times were recorded for each mouse to align the data to the time of injection. Comparisons of mean body temperature after PBS or CNO utilized the 6 hr following injection.

Clozapine-N-oxide (Sigma) was prepared as a 0.1 mg/ml solution in PBS and injected at 1 mg/kg intraperitoneally at ZT14.

### Animals (Sleep)

All procedures were conducted in accordance with NIH guidelines and approved by the Institutional Animal Care and Use Committee of Northwestern University. Opn4Cre and Brn3bz-dta were maintained on a mixed C57Bl/6J; 129Sv/J background (*Hattar et al., 2002*; *Hattar et al., 2006*; *Mu et al., 2005*). Male and female littermate $Opn4^{Cre/+}$ and $Opn4^{Cre/+}$; $Brn3b^{z-dta/+}$ animals between the ages of 2 and 3 months were used for sleep analysis.

### Sleep recording

Male and female littermate $Opn4^{Cre/+}$ and $Opn4^{Cre/+}$; $Brn3b^{z-dta/+}$ mice were used for sleep recordings. Electroencephalogram (EEG) and electromyogram (EMG) electrode implantation was performed simultaneously at 8 weeks of age. Mice were anesthetized with a ketamine/xylazine (98 and10 mg/kg respectively) and a 2-channel EEG and 1-channel EMG implant (Pinnacle Technology) was affixed to the skull. Mice were transferred to the sleep-recording cage 6 days after surgery, tethered with a preamplifier, and allowed 3 days to acclimate to the new cage and tether. Mice were housed in 12:12 light/dark conditions before and after EEG implantation. EEG and EMG recording began simultaneously at the end of the habituation period, which were displayed on a monitor and stored in a computer for analysis of sleep states. The high pass filter setting for both EEG channels was set at 0.5 Hz and low pass filtering was set at 100 Hz. EMG signals were high pass filtered at 10 Hz and subjected to a 100 Hz low pass cutoff. EEG and EMG recordings were collected in PAL 8200 sleep recording software (Pinnacle Technology) and scored, using a previously described, multiple classifier, automatic sleep scoring system, into 10 s epochs as wakefulness, NREM sleep, or REM sleep on the basis of rodent sleep criteria (*Gao et al., 2016*). Light source for all sleep experiments was a 3000 Kelvin light source at 500 lux.

### Wheel-running activity and masking experiment

Mice were placed in cages with a 4.5-inch running wheel, and their activity was monitored with Vital-View software (MiniMitter). Analyses of wheel running activity were calculated with ClockLab (Actimetrics). We used 500 lux light intensity. Mice were initially placed under 12:12 LD masking experiments. Mice were exposed, in their home cage, to a timer-controlled 3 hr light pulse at ZT14-ZT17. Percent activity for each mouse was normalized to its own activity at ZT13 (1 hr before light pulse), and analyzed in 30 min bins.

### Tissue staining and imaging

Animals were anesthetized with Avertin and euthanized prior to fresh dissection of retinas in PBS. Retinas were fixed in 4% paraformaldehyde (Sigma) for at least 1 hr on ice. Retinas were then washed in PBS before staining overnight in anti-OPN4 antibody (1:1000, Advanced Targeting Systems) and then washed prior to 2 hr in secondary antibody (1:1000 goat anti-rabbit AlexaFluor 488, Life Technologies). Retinas were then flat-mounted on slides and imaged on a Zeiss LSM 710 confocal microscope.

### Statistics

All statistical tests were performed in Graphpad Prism or R 3.4.4. Specific tests are listed in the text and figure legends. Linear mixed models were performed with the R packages lme4 1.1–21 and emmeans 1.3.4.

### Data availability

All raw data are linked to this manuscript and available online.

## Additional information

### Funding

| Funder | Grant reference number | Author |
|---|---|---|
| Klingenstein-Simons Fellowship in the Neurosciences | Fellowship in the Neurosciences | Tiffany M. Schmidt |
| Sloan Research Fellowship in Neuroscience | Research Fellowship in Neuroscience | Tiffany M. Schmidt |
| National Institutes of Health | 1DP2EY027983 | Tiffany M. Schmidt |
| National Institutes of Health | GM076430 | Samer Hattar |
| National Institutes of Health | EY024452 | Samer Hattar |

The funders had no role in study design, data collection and interpretation, or the decision to submit the work for publication.

### Author contributions

Alan C Rupp, Conceptualization, Data curation, Formal analysis, Investigation, Visualization, Methodology, Writing—original draft, Project administration, Writing—review and editing; Michelle Ren, Melissa Richardson, Conceptualization, Formal analysis, Investigation, Methodology, Writing—review and editing; Cara M Altimus, Conceptualization, Methodology, Writing—review and editing; Diego C Fernandez, Conceptualization, Investigation, Methodology, Writing—review and editing; Fred Turek, Conceptualization, Supervision, Methodology, Writing—review and editing; Samer Hattar, Tiffany M Schmidt, Conceptualization, Supervision, Funding acquisition, Methodology, Writing—original draft, Project administration, Writing—review and editing

### Author ORCIDs

Alan C Rupp (iD) http://orcid.org/0000-0001-5363-4494
Samer Hattar (iD) https://orcid.org/0000-0002-3124-9525
Tiffany M Schmidt (iD) https://orcid.org/0000-0002-4791-6775

### Ethics

Animal experimentation: This study was performed in accordance with the Institutional Care and Use Committees of Johns Hopkins and Northwestern Universities (IS00003845, IS00000887, IS00000745).

### Decision letter and Author response

Decision letter https://doi.org/10.7554/eLife.44358.023
Author response https://doi.org/10.7554/eLife.44358.024

## Additional files

### Supplementary files

• Transparent reporting form
DOI: https://doi.org/10.7554/eLife.44358.021

### Data availability

All data generated are plotted as individual points on graphs wherever possible and source data files have been provided for Figures 1 to 4.

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
