## [Decision Letter]

Thank you for submitting your article "Distinct ipRGC subpopulations mediate light's acute and circadian effects on body temperature and sleep" for consideration by *eLife*. Your article has been reviewed by three peer reviewers, including Stephen Liberles as the Reviewing Editor and Reviewer #1, and the evaluation has been overseen by Catherine Dulac as the Senior Editor.

The reviewers have discussed the reviews with one another and the Reviewing Editor has drafted this decision to help you prepare a revised submission.

The reviewers were generally positive about the submitted work, but also raised important caveats that would need to be addressed prior to publication. In particular, all three reviewers raised concerns about the specificity of gain-of-function experiments involving DREADDs. Furthermore, reviewer #2 raised an important statistical concern questioning the validity of results involving ablation of Brn3b(+) ipRGCs. Some other information about Brn3b expression, and model validation is also requested. Finally, two of the reviewers thought that additional positive insights into brain regions that may mediate observed effects would broaden the impact and appeal of the paper, but after discussion, it was recognized that this would add significant time for additional experiments, so we leave it only as a suggestion to you for improving your paper rather than as an essential revision. I attached the full reviews below in case they are helpful, and also indicate which requests were considered essential.

Essential revisions:

1) Address questions about the gain-of-function DREADD model used. (See reviewer #1 comment #2, reviewer #2 comment #2, and reviewer #3 comment #5).

2) Address statistical concerns (see reviewer #2, comment #1 and reviewer #3, comment #4).

3) Provide additional information about Brn3b(+) ipRGCs. (See reviewer #1 comment #3)

*Reviewer #1:*

This nice study provides evidence that the acute effects of melanopsin on body temperature and sleep regulation are mediated by ipRGCs that do not target the SCN. The topic would be interesting to the broad readership of *eLife*, and the data appear generally convincing with experiments well performed. This manuscript relies predominantly on loss-of-function and gain-of-function manipulations of ipRGC subpopulations containing Brn3b, and some further characterization of mouse models is needed to support conclusions. Also, the Brn3b population still seems quite broad, so positive identification of brain regions relevant for observed effects would strengthen the paper.

1) Brn3b(+) ipRGCs project to multiple brain regions; evidence in this paper is largely negative for SCN-projecting ipRGCs, but does not provide positive evidence for other brain regions. It seems tractable to address this using the DREADD-based approach presented in Figure 3. For example, Cre-dependent DREADD-encoding AAVs could be injected into various recipient brain regions of Opn4Cre mice. (One caveat of such an experiment is if individual ipRGCs project to multiple target nuclei, but it would help strengthen the case against SCN involvement).

2) Some validation of the mice used in Figure 3 would be helpful; after AAV injection in Brn3b-Cre mice, are DREADDs expressed in other retinal cell types? Are DREADDs expressed centrally in the brain?

3) Some background information about Brn3b expression would be helpful in the Introduction. What% of all ipRGCs express Brn3b, and what% of those that target the SCN and other key nuclei express Brn3b?

*Reviewer #2:*

In this study the authors use intersectional transgenic to determine that ipRGCs contribute to acute changes in body temperature and sleep that are triggered by light. Furthermore, they determine that the population of ipRGCs that mediates these acute change in temperature and sleep is distinct from the population that mediates photoentrainment of circadian rhythms-circadian rhythms instead account for the modulation of body temperature and sleep on a circadian (24 hour) timescale. The evidence to support this conclusion is threefold. First, Opn4ko mice lack acute, light-induced changes in body temperature, while mice lacking rod- or cone-mediated phototransduction retain these acute light-induced changes. This indicates that ipRGCs whose light response is dominated by melanopsin are responsible for mediating this effect. Second, mice that lack the ipRGCs that project to non-SCN targets in the brain also lack acute-light-induced changes in body temperature. Note, in this case, the ipRGCs are killed by diptheriatoxin. Third, activation of *all* Brn3b(+) RGCs (which is most of them), including Brn3b(+) ipRGCs, via DREADDs induced a drop in body temperature. Finally, the authors show that killing Brn3b(+) ipRGCs prevents acute light-induced sleep, though it is not clear whether this is via the same pathway as acute light-induced changes in temperature.

The observation that different ipRGC-subtypes mediate photoentrainment of circadian rhythm vs. light-induced acute changes in sleep and body temperature through distinct circuits is significant. The data are clearly presented. However, I have a major concern about the primary effect as described in comment #1 below. Comment #2 is also important but more easily addressed.

1) Figure 2G: The authors argue that there is no acute change in body temperature in mice lacking Brn3b(+) ipRGCs. However, 5/7 mice still have a temperature decrease that is comparable in size to the temperature decrease in control mice. Therefore, the correct comparison to make to test their hypothesis is between the change in body temperature in the Opn4^Cre/+^ Brn3b^DTA/+^ mice vs. the Opn4^Cre/+^ (control) mice. This is a major comment that needs to be addressed by the authors.

2) Figure 3: For the DREADD experiment, hM3Dq is expressed in all RGCs that are Brn3b(+). Hence, the induction of temperature changes could be via an entirely different pathway, potentially involving the dLGN. For this reason, it would be important for the authors to state exactly how much transfection they saw and what percent of ipRGCs express the DREAAD. For example, the authors should explain whether their 50% transfection criterion is 50% of ipRGCs or of all RGCs? (subsection “Body temperature recordings”, second paragraph). The authors need to include this caveat in the discussion of this result. In addition, the example image is not large enough for the reader to draw any conclusion about the expression of hM3Dq, except that it is widespread. A larger image and annotations where overlap exists between anti-Opn4 antibody and mCherry would help the interpretation of this figure.

*Reviewer #3:*

The authors investigate the contribution of melanopsin expressing intrinsically photosensitive ganglion cells to two aspects of photo-regulated behavior- temperature and sleep. They conduct genetic and functional experiments in an attempt to demonstrate that Brn3b(+)melanopsin RGCs are responsible for 'acute' responses to a light pulse. There is much to like about this paper, including the compelling nature of the question. The paper is also well written. However, there are several issues with the manuscript that result in a somewhat lukewarm view of the paper in its current form for *eLife*. These include the following:

1) Scientific claims of the paper appear overstated. In the Abstract the authors write that "body temperature and sleep responses to light are absent after genetic ablation of all ipRGC." Given this, one would expect dramatically reduced responses to light. Instead, nearly all light induced temperature and sleep responses are intact in each of the mutant genotypes analyzed. The difference that are observed are relatively small and restricted to turning on the lights for 3 hours at night. The title also claims that "distinct ipRGC subpopulations" regulate light's acute affect. The Brn3b-DTA model they use ablates nearly all melanopsin (likely upwards of ~90%, Discussion). To call this a distinct subset seems a bit problematic. This leads to issue two:

2) Somewhat incremental advance in the field. The primary news of the manuscript seems to be that melanopsin is not required for the majority of body temperature regulation by light (leaving one wondering what is) but only for 'acute' responses when light is turned on at a specific point in the dark cycle. This finding, while interesting, does not dramatically move the field forward.

3) Sparse data. Overall, the depth of the experiments and statistical rigor leaves some room for improvement. There are many more experiments the authors themselves mention that would greatly improve the quality of the manuscript. Chief among these are data on true melanopsin ipRGC subtypes and their role in these responses. In parallel the authors do not address the role of particular brain regions in these behaviors. While these studies are difficult, they would greatly elevate this paper. But several other simpler experiments would also help. These include: (1) repeating the studies using only wavelengths of light to which only melanopsin responds to rule out effects from other opsins (479nm or blue light); (2) performing tracing experiments to the brain from single ipRGCs; and (3) a more complete study of the 'acute' light responses which could include examining the effects of a shorter light burst.

4) Lack of appropriate statistical comparisons. The key conclusions of the central figures rely on differences between changes in body temperature between control and experimental groups. While the authors statistically compare light and dark temperatures within each group, they do not appear to compare between them in Figures 1, 2, or 3. This would be required to validate the central claims of the figures and includes statistically comparisons of data in Figure 1G and I, Figure 2D and F, and Figure 3D and F.

5) Problematic gain of function experiments. In Figure 3, the authors attempt to show that activation of Brn3b(+) melanopsin+ RGCs induces temperature decreases. However, from the image shown, less than ~10% of OPN4+ ipRGCs are transduced, and even more problematic, ~90% of the transduced RGCs are not OPN4+ ipRGCs. Thus, the main conclusion we can draw is that RGC activation broadly induces a temperature decrease. This experiment would need to be repeated using the OPN4Cre driver that is featured in the other figures.

[Editors' note: further revisions were requested prior to acceptance, as described below.]

Thank you for submitting your article "Distinct ipRGC subpopulations mediate light's acute and circadian effects on body temperature and sleep" for consideration by *eLife*. Your article has been reviewed by one peer reviewer (prior reviewer #2), and the evaluation has been overseen by a Reviewing Editor and Catherine Dulac as the Senior Editor. The reviewer has opted to remain anonymous.

The Reviewing Editor has drafted this decision to help you prepare a revised submission.

A major concern of reviewer #2 persists (please see comment #1 below). Perhaps repeating the experiment with a new cohort of mice would clarify. Also, the new supplementary figure (Figure 3—figure supplement 1D) seems to indicate that CNO exerts an effect in control mice (comment #3 below), an issue which needs to be addressed. Finally, please note other comments related to statistical comparisons (points #4,5).

*Reviewer #2:*

1) The primary concern regarding the data in Figure 2 is not fully addressed. The authors present the data as before and it still appears that 5/7 Brn3b-DTA mice had a temperature decrease comparable to that of control mice. Also, two control mice have a temperature difference that is less than that of the Brn3b-DTA mice.

The authors did add a new supplementary figure (Figure 2—figure supplement 1) that would directly compare the temperature changes across genotypes and resolve our concern. However, if these are the same mice that are plotted in Figure 2, then some mistake has been made or the data are analyzed in a different way that is not clearly explained. What the reader expects is for the changes in body temperature from the dark to light condition portrayed in Figure 2E (Control mice) to be compared to those from Figure 2G (Brn3b-DTA mice) but the body temperature changes in Figure 2—figure supplement 1 appear to originate from a different data set. For example, Figure 2—figure supplement 1 portrays two Control mice with a temperature decrease of more than 1 degree, but there is no Control mouse in Figure 2E that exhibited a -1 degree change in temperature, let alone a -1.5 degree change. The authors need to look at their data closely – either what is plotted in Figure 2 or what is plotted in Figure 2—figure supplement 1 is incorrect.

2) The authors have provided a quantification of the RGCs that express the DREAADs for the gain-of-function experiment in Figure 3. This is important to provide, and it appears that 50% of all RGCs are expressing the DREAAD. The authors make the argument that the DREAADs expressed in ipRGCs are likely the only impactful ones based on their finding that melanopsin expression by ipRGCs is required for the change in body temperature evoked by a light pulse during subjective night and that ipRGCs are on their own sufficient to evoke the change in body temperature in the absence of rod and cone inputs to the retina. The authors also provide a good discussion of the potential caveats of this approach. Though the experiment is not perfect, making it perfect would require development of new technology. I am satisfied with the current description of results.

3) It appears in Figure 3—figure supplement 1 that CNO evokes a small decrease in body temperature (~0.5 degrees) in mice that do not express the receptor hM3D(G_q_) (control mice). This effect is in the same direction and of similar magnitude to the effect of CNO injection in mice that do express the receptor (Brn3b-Cre mice). The authors need to compare these two populations directly rather than compare each population to the PBS control. This will rule out an effect on body temperature coming from off-target effects of CNO.

4) The figure legends should state whether the error shading around the body temperature traces is SEM or SD, starting with Figure 1.

5) In several places, linear mixed models are used in the context of performing statistical comparisons (Figure 1—figure supplement 2, Figure 2—figure supplement 1, etc.). Some details should be provided in the Materials and methods section to describe how these linear models work.

Note, these last two points are both getting at the same point. The authors indicate in their response that the effects are big and reliable. Indeed many of the example traces indicate that is the case. Yet somehow this large, reliable effect is not translating into their summary data, which appears to be highly variable. In addition, there appears to be a distinct temperature change during the light pulse in subjective night in the Opn4 KO mouse (Figure 1), yet the authors describe it as being "absent". Either is fine – large and reliable, or highly variable, but the description of the effect needs to be consistent.

---

## [Author Response]

Essential revisions:1) Address questions about the gain-of-function DREADD model used. (See reviewer #1 comment #2, reviewer #2 comment #2, and reviewer #3 comment #5).

Before directly addressing the DREADD experiments and associated revisions, we would like to highlight the evidence (in this and other papers) demonstrating that ipRGCs are required for acute thermoregulation and sleep induction by light. First, we find that lack of melanopsin results in loss of changes in body temperature to a light pulse at ZT14-16 (Figure 1), as does ablation of Brn3b(+) ipRGCs (Figure 2). Second, previous studies have demonstrated that both loss of melanopsin expression and ablation of ipRGCs results in a loss of acute light induction of sleep, demonstrating that ipRGCs are necessary for this behavior (Altimus et al., 2008). Thus, these data indicate that ipRGCs are necessary for the acute light-induced changes in body temperature and sleep, making it likely that activation of Brn3b(+) RGCs (Figure 3) is driving effects on body temperature through ipRGCs. Nonetheless, as discussed below, we do provide an expanded discussion of the important caveats of this manipulation.

2) Address statistical concerns (see reviewer #2, comment #1 and reviewer #3, comment #4).3) Provide additional information about Brn3b+ ipRGCs. (See reviewer #1 comment #3)Reviewer #1:[…]1) Brn3b(+) ipRGCs project to multiple brain regions; evidence in this paper is largely negative for SCN-projecting ipRGCs, but does not provide positive evidence for other brain regions. It seems tractable to address this using the DREADD-based approach presented in Figure 3. For example, Cre-dependent DREADD-encoding AAVs could be injected into various recipient brain regions of Opn4Cre mice. (One caveat of such an experiment is if individual ipRGCs project to multiple target nuclei, but it would help strengthen the case against SCN involvement).

We agree that understanding which brain regions drive these behaviors is an incredibly important question. There are multiple potential candidates that we have outlined in the Discussion, and we are actively following up on these questions in separate studies. We agree with the consensus reached by the reviewers after discussion that this would be beyond the scope of the current study.

2) Some validation of the mice used in Figure 3 would be helpful; after AAV injection in Brn3b-Cre mice, are DREADDs expressed in other retinal cell types? Are DREADDs expressed centrally in the brain?

We have performed additional quantification of the overlap between the AAV reporter and Opn4+ cells. We have also quantified the fraction of Opn4+ cells that are transduced. Overall, the proportion of Opn4+ ipRGCs that are show detectable mCherry fluorescence is ~40%, which is well in line with what we would expect: M1 and M2 ipRGCs represent the vast majority of Opn4+ immunopositive RGCs. M2 ipRGCs represent up to 50% of Opn4+ RGCs and are all Brn3b(+) (Chen et al., 2011). M1 ipRGCs represent at least 50% of Opn4+ RGCs, but just 15% of adult M1 ipRGCs retain Brn3b expression beyond development (Chen et al., 2011). Thus, the maximum possible expression levels we would expect is ~60% of Opn4+ RGCs that label with the reporter. Coupled with the unavoidable fact that some cells will simply not be infected with the AAV, these expression patterns are aligned with our expectations. Additionally, the vast majority of DREADD-expressing RGCs in the retina are not Opn4+. This is expected given that 80% of all RGCs are Brn3b(+). However, as mentioned above, we think it unlikely (though of course not impossible) that these non-ipRGCs drive the associated temperature changes. These data are now included as Figure 3—figure supplement 1.

DREADD expression is confined to the eye because we injected the AAVs directly into the eye. We have never seen evidence of AAV infection in the brain using this experimental paradigm (Takuma Sonoda, Unpublished Data) as many other laboratories who carry out such experiments. Therefore, we are confident that the expression is restricted to the eye.

3) Some background information about Brn3b expression would be helpful in the Introduction. What% of all ipRGCs express Brn3b, and what% of those that target the SCN and other key nuclei express Brn3b?

We thank the reviewer for suggesting additional information be added about Brn3b(+) ipRGCs. We have added additional, and more explicit, explanation to the Results. This enhanced description now better describes the Brn3b(+) and Brn3b(–) ipRGC populations, the subtypes that fall into each category, and their known behavioral functions. We chose to add it in the Results rather than the Introduction because that is where we begin manipulating ipRGCs based on their expression of Brn3b.

Reviewer #2:[…] The observation that different ipRGC-subtypes mediate photoentrainment of circadian rhythm vs. light-induced acute changes in sleep and body temperature through distinct circuits is significant. The data are clearly presented. However, I have a major concern about the primary effect as described in comment #1 below. Comment #2 is also important but more easily addressed.1) Figure 2G: The authors argue that there is no acute change in body temperature in mice lacking Brn3b(+) ipRGCs. However, 5/7 mice still have a temperature decrease that is comparable in size to the temperature decrease in control mice. Therefore, the correct comparison to make to test their hypothesis is between the change in body temperature in the Opn4^Cre/+^ Brn3b^DTA^ mice vs. the Opn4^Cre/+^ (control) mice. This is a major comment that needs to be addressed by the authors.

We thank both reviewer #2 and reviewer #3 for raising this important point. Most importantly for the central conclusions of this paper: we have now compared the change in body temperature in darkness versus light pulse both within Brn3b^DTA/+^ animals and between Control and Brn3b^DTA/+^ animals. In support of our initial conclusions, we find no significant change in body temperature in Brn3b^DTA/+^ relative to the dark condition. We also find that the Cre/+ shows a significantly larger change in body temperature in response to a light pulse compared to Brn3b^DTA/+^. We have plotted these changes as well as the change from baseline in both conditions in Figure 2—figure supplement 1.

As requested, we performed similar comparisons for all genotypes used for thermoregulation and sleep experiments in the paper, and are happy to report that the results are each consistent with our initial interpretations. The results can be found in the following figures: Figure 1—figure supplement 2, Figure 2—figure supplement 1, Figure 3—figure supplement 1, and Figure 4—figure supplement 2. We thank the reviewers for suggesting these analyses as they greatly strengthen our conclusions.

2) Figure 3: For the DREADD experiment, hM3Dq is expressed in all RGCs that are Brn3b(+). Hence, the induction of temperature changes could be via an entirely different pathway, potentially involving the dLGN. For this reason, it would be important for the authors to state exactly how much transfection they saw and what percent of ipRGCs express the DREAAD.

We thank the reviewer for highlighting this important point. Though we think it unlikely that a distinct pathway is involved (given the two points mentioned above regarding the necessity of ipRGCs for acute thermoregulation by light), we now quantify the number of cells that are Brn3b(+), Opn4(–); Brn3b(+), Opn4(+), and Brn3b(–), Opn4(+) as well as the proportion of Opn4 immunopositive cells that express G_q_-DREADDs. Though ipRGCs were sometimes more faintly labeled than other RGCs, we were still able detect some mCherry expression in about 40% of Opn4+ cells. These are now identified in Figure 3—figure supplement 1 with arrows.

For example, the authors should explain whether their 50% transfection criterion is 50% of ipRGCs or of all RGCs? (subsection “Body temperature recordings”, second paragraph).

We have added a clarification to this statement (50% of all RGCs assessed by fluorescence detectable across more than half of the retina).

The authors need to include this caveat in the discussion of this result.

This is an important point to highlight for the general readership of *eLife*, and we have added additional discussion of the caveats of this experiment, and thank the reviewer for raising these important points.

As mentioned earlier, we find that lack of melanopsin results in loss of changes in body temperature to a light pulse at ZT14-16 (Figure 1), as does ablation of Brn3b(+) ipRGCs (Figure 2). Moreover, previous studies have demonstrated that both loss of melanopsin expression and ablation of ipRGCs results in a loss of acute light induction of sleep, demonstrating that ipRGCs are necessary for this behavior (Altimus et al., 2008). Thus, these data indicate that ipRGCs are necessary for the acute light-induced changes in body temperature. We therefore chose to next activate Brn3b(+) RGCs because if ipRGCs are in fact required for acute thermoregulation by a light pulse at ZT14-16, then we would expect activating these same cells to mimic the effects of light and cause a subsequent decrease in body temperature. Our finding that activation of Brn3b(+) RGCs causes a reduction in body temperature at ZT14 are therefore consistent with our model. However, we are mindful of the fact that we are activating many non-ipRGCs with this manipulation. When taken in conjunction with our results from Figures 1 and 2 showing ipRGCs are necessary for this behavior, we think it unlikely that a separate pathway is driving the temperature decrease in Figure 3.

In addition, the example image is not large enough for the reader to draw any conclusion about the expression of hM3Dq, except that it is widespread. A larger image and annotations where overlap exists between anti-Opn4 antibody and mCherry would help the interpretation of this figure.

We have added additional examples with images in Figure 3—figure supplement 1.

Reviewer #3:[…] There are several issues with the manuscript that result in a somewhat lukewarm view of the paper in its current form for eLife. These include the following:1) Scientific claims of the paper appear overstated. In the Abstract the authors write that "body temperature and sleep responses to light are absent after genetic ablation of all ipRGC." Given this, one would expect dramatically reduced responses to light. Instead, nearly all light induced temperature and sleep responses are intact in each of the mutant genotypes analyzed. The difference that are observed are relatively small and restricted to turning on the lights for 3 hours at night. The title also claims that "distinct ipRGC subpopulations" regulate light's acute affect. The Brn3b-DTA model they use ablates nearly all melanopsin (likely upwards of ~90%, Discussion). To call this a distinct subset seems a bit problematic.

We agree with the reviewer that we are ablating a large number of ipRGCs in the Brn3bDTA animal. However, it is the combination of identifying a subset of ipRGCs that is required for these behaviors in combination with the fact that the light input for these acute, light-evoked behaviors does not appear to be routed through the SCN as previously believed. This is a significant departure from the current models.

This leads to issue two:2) Somewhat incremental advance in the field. The primary news of the manuscript seems to be that melanopsin is not required for the majority of body temperature regulation by light (leaving one wondering what is) but only for 'acute' responses when light is turned on at a specific point in the dark cycle. This finding, while interesting, does not dramatically move the field forward.

By acute, we mean that the light pulse is relatively short term and not recurring in a cyclic manner. Relative to the 12 hour light/dark cycle, the single, 3 hour stimulus is acute. This type of stimulus is meant to mimic unexpected encounters with environmental light out of sync with the environmental light/dark cycle. As mentioned above, it was previously believed that light input from the retina to the SCN drove both the circadian and acute responses to environmental light. Our data show that this is not the case. Thus, we have shown that light input to the SCN alone is not sufficient for light’s acute effects, and attributed these effects to a specific subset of ipRGCs. These findings have important implications for understanding how light information is integrated over different timescales, and identifies a subset of RGCs to target for future studies.

3) Sparse data. Overall, the depth of the experiments and statistical rigor leaves some room for improvement. There are many more experiments the authors themselves mention that would greatly improve the quality of the manuscript. Chief among these are data on true melanopsin ipRGC subtypes and their role in these responses. In parallel the authors do not address the role of particular brain regions in these behaviors. While these studies are difficult, they would greatly elevate this paper. But several other simpler experiments would also help. These include: (1) repeating the studies using only wavelengths of light to which only melanopsin responds to rule out effects from other opsins (479nm or blue light); (2) performing tracing experiments to the brain from single ipRGCs; and (3) a more complete study of the 'acute' light responses which could include examining the effects of a shorter light burst.

1) Regarding the influence of other options: Our data clearly show that absence of melanopsin (Figure 1) leads to a severe deficit in light-evoked temperature decreases to a 3 hour light pulse at ZT14-16, ruling out a sufficiency of other opsins in driving this behavior. Thus, we feel that the use of additional wavelengths would be somewhat redundant.

2) We agree with the author that understanding the connectivity of the circuit will be very important in the future, but we do not yet have a means to label single RGC axons arising from a defined population, and have not identified the pertinent brain regions for thermoregulation and sleep by light as of yet.

3) We chose this three hour light pulse given its standard use as a “masking” pulse in previous studies. This allowed us to compare more directly with previous studies of circadian photoentrainment, negative masking, and sleep induction by light (Mrosovsky et al., 1999; Mrosovsky and Hattar, 2003; Altimus et al., 2008; Lupi et al., 2008). We agree that there are many interesting experiments that could be done to test the sensitivity of the system to light stimuli of varying lengths and intensities.

4) Lack of appropriate statistical comparisons. The key conclusions of the central figures rely on differences between changes in body temperature between control and experimental groups. While the authors statistically compare light and dark temperatures within each group, they do not appear to compare between them in Figures 1, 2, or 3. This would be required to validate the central claims of the figures and includes statistically comparisons of data in Figure 1G and I, Figure 2D and F, and Figure 3D and F.

We thank the reviewer for this suggestion. Please see our response to reviewer #2, comment #1.

5) Problematic gain of function experiments. In Figure 3, the authors attempt to show that activation of Brn3b+ melanopsin+ RGCs induces temperature decreases. However, from the image shown, less than ~10% of OPN4+ ipRGCs are transduced, and even more problematic, ~90% of the transduced RGCs are not OPN4+ ipRGCs. Thus, the main conclusion we can draw is that RGC activation broadly induces a temperature decrease. This experiment would need to be repeated using the OPN4cre driver that is featured in the other figures.

The reviewer raises important points. As mentioned in our responses to reviewer #1, comment 2 and reviewer #2, comment #2, we have now provided additional quantification of the infection rate as well as added additional discussion of the caveats of this experiment. We have chosen not to use Opn4^Cre^ as a driver because it does not get at the central question of this paper, which is whether there are separate circuits for acute versus circadian effects of light. Activation of all ipRGCs would not allow us to differentiate between the circuits underlying these two effects because we would also be activating Brn3b-negative ipRGCs, which we know are sufficient for circadian photoentrainment of body temperature to light/dark cycles. Therefore, it would be difficult to interpret these experiments, regardless of the outcome.

[Editors' note: further revisions were requested prior to acceptance, as described below.]

Reviewer #2:1) The primary concern regarding the data in Figure 2 is not fully addressed. The authors present the data as before and it still appears that 5/7 Brn3b-DTA mice had a temperature decrease comparable to that of control mice. Also, two control mice have a temperature difference that is less than that of the Brn3b-DTA mice.The authors did add a new supplementary figure (Figure 2—figure supplement 1) that would directly compare the temperature changes across genotypes and resolve our concern. However, if these are the same mice that are plotted in Figure 2, then some mistake has been made or the data are analyzed in a different way that is not clearly explained. What the reader expects is for the changes in body temperature from the dark to light condition portrayed in Figure 2E (Control mice) to be compared to those from Figure 2G (Brn3b-DTA mice) but the body temperature changes in Figure 2—figure supplement 1 appear to originate from a different data set. For example, Figure 2—figure supplement 1 portrays two Control mice with a temperature decrease of more than 1 degree, but there is no Control mouse in Figure 2E that exhibited a -1 degree change in temperature, let alone a -1.5 degree change. The authors need to look at their data closely-either what is plotted in Figure 2 or what is plotted in Figure 2—figure supplement 1 is incorrect.

We now realize that we did not clearly explain our new analysis, leading to confusion about the data and analyses. Throughout the paper we have performed two distinct analyses on the same dataset. In the original submission and in the data plotted in Figure 1H, J and Figure 2E, G, we have compared the mean of the absolute body temperature on the “Dark” (control) night from ZT14-17 to the mean of the absolutebody temperature on the “Light” night from ZT14-17 (see Author response image 1). If light induces a decrease in body temperature, then we should see significantly lower temperatures in the “Light” condition than in the “Dark” condition for each genotype. We did not see a significant decrease in melanopsin null animals (Figure 1J) or in Brn3b-DTA animals (Figure 2G, also shown in Author response image 1). The reviewers noted that the mean in the “Light” compared to the mean in the “Dark” across the two nights is lower for some Brn3b-DTA animals, but this did not reach statistical significance for the population.

An excellent suggestion from the reviewers for the second submission to ensure that there was no actual light evokedchange in body temperature given the trends mentioned above, was to more directly measure temperature changes *in response to the light pulse* instead of comparing the same time frame across days. Therefore, we performed new analyses of all of the data, which are represented in Figure 1—figure supplement 2, Figure 2—figure supplement 1, and Figure 3—figure supplement 1. In these analyses, we performed a completely distinct set of comparisons where we compared the *change* in body temperature of mice on the “Control” (Dark night) from ZT14 (which served as a baseline temperature) to the mean body temp during the entire time period of ZT14-17. We then compared that change to the *change* in body temperature on the “Light” night between the ZT14 baseline and the time period of ZT14-17 during which the mice were exposed to a light pulse (see Author response image 1). This allowed us to directly measure changes in body temperature from the baseline temperature in response to the light pulse. In this case, if light causes a decrease in body temperature then we would expect the *change* in body temperature in “Light” to be significantly different from the change in the “Ctrl” condition within a genotype. This analysis allows us to compare how the temperature actually changed within the same animal, on the same night, in response to the light pulse itself. This method accounts for any light-independent changes in body temperature due to, for example, daily variation. These new analyses supported the same conclusions as those from the first submission: there was no change in body temperature in either melanopsin null or Brn3b-DTA animals. As an example, we have included the plots for the Control versus Brn3b-DTA genotypes from Figure 2—figure supplement 1 in Author response image 1. Here you can see that for the Brn3b-DTA animals, many animals showed a similar decrease in body temperature compared to their temperature at ZT14 on *both* the Ctrl and Light nights. In other words, there was *no further decrease* in body temperature in the presence of light, arguing against a light-induced change in body temperature, and supporting our original conclusions for not only Brn3b-DTA animals (Figure 2—figure supplement 1), but also melanopsin null animals (Figure 1—figure supplement 2). Because of the different comparisons, the data points from the two analyses we performed (Mean Body Temperature versus Mean Change in Body Temperature) cannot be directly compared and will not give identical amplitudes, which we believe was the source of confusion for the reviewer. We apologize for the confusion and have attempted to clarify this in the figure legends. Importantly, these data for not only Brn3b-DTA, but also each of our other mutant lines, are in complete support of our original analyses, and we thank the reviewers for suggesting these additional comparisons. We have provided a schematic of the comparisons in Author response image 1.

**Author response image 1. respfig1:** Schematic of analyses throughout the manuscript with example graphs from Brn3bDTA analyses.

Regarding the reviewer’s observations of temperature decreases in Brn3b^DTA^ animals: we hope that our new explanation clarifies the confusion. Brn3b-DTA mice did not display body temperature reductions similar to Controls because Figure 2—figure supplement 1 shows that the drop in body temperature in Brn3b-DTA mice on the “Light” night is *completely similar* in magnitude to their normal change in body temperature relative to ZT14 on the “Control” night where no light pulse was given (and also match the normal changes observed on the “Control” night in control animals). Additionally, no Brn3b^DTA^ animals reach the average value of the control animal body temperature drop during the “Light.” We thank the reviewers for suggesting this new analysis because it allowed us to more directly compare natural temperature variations to those induced by light on the same night, and more strongly supports our original conclusions. We feel that the additional clarifications for the measurement methods in the figure captions will help clarify this for readers.

Importantly, we have included all raw data with the submission of this manuscript, so interested readers will be able to investigate changes in individual animals if they should wish.

2) The authors have provided a quantification of the RGCs that express the DREAADs for the gain-of-function experiment in Figure 3. This is important to provide, and it appears that 50% of all RGCs are expressing the DREAAD. The authors make the argument that the DREAADs expressed in ipRGCs are likely the only impactful ones based on their finding that melanopsin expression by ipRGCs is required for the change in body temperature evoked by a light pulse during subjective night and that ipRGCs are on their own sufficient to evoke the change in body temperature in the absence of rod and cone inputs to the retina. The authors also provide a good discussion of the potential caveats of this approach. Though the experiment is not perfect, making it perfect would require development of new technology. I am satisfied with the current description of results.

We are happy that the reviewer is satisfied with the additional quantification and discussion of this approach.

3) It appears in Figure 3—figure supplement 1 that CNO evokes a small decrease in body temperature (~0.5 degrees) in mice that do not express the receptor hM3D(G_q_) (control mice). This effect is in the same direction and of similar magnitude to the effect of CNO injection in mice that do express the receptor (Brn3b-Cre mice). The authors need to compare these two populations directly rather than compare each population to the PBS control. This will rule out an effect on body temperature coming from off-target effects of CNO.

In order to test for significant effects of CNO injection in the absence of DREADDs, we feel that the appropriate comparison is to perform a within animal comparison of the effects of PBS injection in an animal versus CNO injection within that same animal, which we have done for 9 animals in the Control group (i.e. each animal received an injection of PBS and an injection of CNO on separate occasions). This allows us to compare the effect of these two different compounds on the same animal. When we perform this experiment, we see no significant change in body temperature, though we see the trend that the reviewer is referring to. Importantly, when we perform these same within animal comparisons in Brn3bCre-G_q_DREADD mice, we do see a statistical difference in CNO versus PBS injection across 8 animals.

Of note regarding statistical power to compare CNO across the two genotypes: Our inability to detect statistically different effect in Brn3b + CNO and Control + CNO is due to this lack of pairing; our power analyses indicate that we would need to run > 40 mice per group to detect a difference given the magnitude and variance of this dataset.

Therefore, while the statistical model cannot rule out the possibility that CNO has a small effect on body temperature, we believe reasonable interpretation suggests an effect of Brn3b(+) RGC activation on body temperature. CNO injection in Brn3b-G_q_DREADD mice drives a robust reduction in body temperature that remains ~1°C lower than at injection time for ≥ 6 hours (Figure 3D). There does seem to be a drop in body temperature following CNO injection in Control mice, but it is roughly comparable to that seen with PBS injection and is within ~0.2°C of baseline within a few hours (Figure 3—figure supplement 2C). The difference in magnitude of the effect is borne out in the summary data as well, with approximate doubling of the average body temperature reduction for the 6 hours after injection in Brn3b-Cre vs. Control with CNO (Figure 3—figure supplement 1D).

4) The figure legends should state whether the error shading around the body temperature traces is SEM or SD, starting with Figure 1.

We apologize for this oversight. The figures have now been labeled with the appropriate variance measurement.

5) In several places, linear mixed models are used in the context of performing statistical comparisons (Figure 1—figure supplement 2, Figure 2—figure supplement 1, etc.). Some details should be provided in the Materials and methods section to describe how these linear models work.

We apologize that the details of the linear mixed models were not included in the Materials and methods. Linear mixed models are a statistical alternative to ANOVA models. They are preferable in instances of repeated measures because they are capable of dealing with missing values and of incorporating continuous covariates (such as time, light intensity, etc). All linear mixed models were carried out using the R packages lme4 1.1-21 and emmeans 1.3.4. These packages are now explicitly listed in the Materials and methods.

Note, these last two points are both getting at the same point. The authors indicate in their response that the effects are big and reliable. Indeed many of the example traces indicate that is the case. Yet somehow this large, reliable effect is not translating into their summary data, which appears to be highly variable. In addition, there appears to be a distinct temperature change during the light pulse in subjective night in the Opn4 KO mouse (Figure 1), yet the authors describe it as being "absent". Either is fine – large and reliable, or highly variable, but the description of the effect needs to be consistent.